# Binary-Phase vs. Frequency Modulated Radar Measured Performances for Automotive Applications

**DOI:** 10.3390/s23115271

**Published:** 2023-06-01

**Authors:** Mattia Caffa, Francesco Biletta, Riccardo Maggiora

**Affiliations:** Department of Electronics and Telecommunications, Politecnico di Torino, 10129 Torino, Italy; francesco.biletta@polito.it (F.B.); riccardo.maggiora@polito.it (R.M.)

**Keywords:** automotive radar, PMCW, FMCW, millimeter wave radars, radar waveforms

## Abstract

Radars have been widely deployed in cars in recent years, for advanced driving assistance systems. The most popular and studied modulated waveform for automotive radar is the frequency-modulated continuous wave (FMCW), due to FMCW radar technology’s ease of implementation and low power consumption. However, FMCW radars have several limitations, such as low interference resilience, range-Doppler coupling, limited maximum velocity with time-division multiplexing (TDM), and high-range sidelobes that reduce high-contrast resolution (HCR). These issues can be tackled by adopting other modulated waveforms. The most interesting modulated waveform for automotive radar, which has been the focus of research in recent years, is the phase-modulated continuous wave (PMCW): this modulated waveform has a better HCR, allows large maximum velocity, permits interference mitigation, thanks to codes orthogonality, and eases integration of communication and sensing. Despite the growing interest in PMCW technology, and while simulations have been extensively performed to analyze and compare its performance to FMCW, there are still only limited real-world measured data available for automotive applications. In this paper, the realization of a 1 Tx/1 Rx binary PMCW radar, assembled with connectorized modules and an FPGA, is presented. Its captured data were compared to the captured data of an off-the-shelf system-on-chip (SoC) FMCW radar. The radar processing firmware of both radars were fully developed and optimized for the tests. The measured performances in real-world conditions showed that PMCW radars manifest better behavior than FMCW radars, regarding the above-mentioned issues. Our analysis demonstrates that PMCW radars can be successfully adopted by future automotive radars.

## 1. Introduction

FMCW is the most widespread modulated waveform scheme for radar systems in the automotive industry. It has been adopted by many manufacturers, for its high accuracy and reliability in detecting targets, as well as its low power consumption and compact design. However, binary PMCW radar is gaining popularity in the automotive radar industry, due to its advantages. In fact, binary PMCW radar offers several benefits over FMCW radar, including no range-Doppler coupling, better resilience to interference, larger maximum velocity, and improved HCR. HCR is the radar ability to discriminate between small targets that are in close proximity to large targets, in range and angle [1]. Another advantage of PMCW radar is its ability to exploit joint communication and radar sensing (JCRS), which is also useful for further mitigating mutual interference [2,3] and facilitating ghost targets removal [4]. Comparison of the measured data of binary PMCW and FMCW radar has provided deep insights into their performances. Despite the growing interest in PMCW technology, while many simulations have been extensively performed to analyze and compare its performance to FMCW radar, very few real-world measured data have been shown to support those results.

In [5] a 24 GHz 4 × 4 multiple–input multiple–output (MIMO) PMCW radar is modeled and analyzed and relative simulated results are presented.In [6] a study on the mutual interference between FMCW and PMCW radar shows that all possible victim-interferer pairs (FMCW-FMCW, PMCW-PMCW, PMCW-FMCW, FMCW-PMCW) have similar sensitivity to interference and that a certain degree of randomness in waveforms is required to minimize the probability of interference. A similar study is held in [7] where the different ripple structures that arise in the interference cases are discussed and analyzed. In [8] a 4 × 4 MIMO PMCW radar and different techniques to achieve orthogonality of the signal are presented. Laboratory tests and simulations results are then discussed showing PMCW radars are an attractive technology for automotive scenarios.

Few companies, such as Uhnder and IMEC, have been at the forefront of developing PMCW radars with binary modulation. Uhnder has developed the s80 radar-on-chip (RoC) [9], today the most advanced PMCW radar chip on the market, with 12 Tx/16 Rx channels, internal DSPs and CPUs for data processing, and the ability to eliminate mutual interference, thanks to the proper choice of modulation sequences. IMEC has developed a less ambitious, but nonetheless interesting, PMCW radar chip with 2 Tx/2 Rx, which can be cascaded (two chips) to obtain a 4 Tx/4 Rx MIMO array: in addition, in this case, the radar processing can be performed directly on the chip. The development of the prototype, with full hardware description and some results obtained using the chips, is shown in [8,10].

The present paper aimed to provide a comparison between the raw data of a 1 Tx/1 Rx laboratory-assembled binary PMCW radar and those of an FMCW radar. We discuss the advantages and limitations of each technology. For the binary PMCW radar, a prototype solution was assembled in our laboratory, using connectorized components and an FPGA for real-time processing. For the FMCW radar, an off-the-shelf development board, equipped with a Texas Instruments AWR1843 SoC [11], was adopted. Both firmwares for radar processing were fully developed and implemented on the FPGA and on the SoC, while the software for data visualizations and post-processing was implemented on a laptop PC.

The radars’ parameters and gains were tuned to have the same power budgets, and in such a way that the signal-to-noise ratio (SNR) comparison was not affected by the hardware. A test campaign was conducted, to analyze the range-Doppler responses of the two radars, with real data captured in defined real-world scenarios. For each scenario, the data were captured simultaneously, with the radars close to each other, and in the same conditions, to obtain consistent measurements. The results are compared and discussed in this paper.

The structure of this paper is as follows: in Section 2, the FMCW and binary PMCW radars are described; in Section 3, the description of the experimental setup is presented; in Section 4, the main processing steps of the two radars, and the key radar parameters, are outlined; in Section 5, a comparative analysis of the measured results captured by the radars in real-world scenarios is presented; in Section 6, conclusions are drawn.

## 2. FMCW and Binary-PMCW Radar Signals Description

A brief description of the modulated waveform and signal processing required for FMCW and PMCW radar is reported below. A well described and more in depth analysys of these and other modulated waveforms can be found in [12].

### 2.1. FMCW Radar

FMCW is the conventional modulated waveform used in automotive radar which has been well analyzed and adopted by manufacturers in the last decades. An FMCW radar periodically transmits a certain number of frequency modulated signals called chirps. The transmitted signal can be modeled as,
(1)$$s_{Tx}\left(t\right)=exp\left(j2\pi f_{0}t+j\pi \frac{B}{T_{c}}t^{2}\right)$$
where $f_{0}$ is the carrier frequency, *B* is the bandwidth and $T_{c}$ is the duration of each chirp. The ideal received scattered signal by one target after a propagation delay $t_{d}=2R/c$ (with *R* being the target range and *c* the speed of light) is,
(2)$$s_{Rx}\left(t\right)=As_{Tx}\left(t-t_{d}\right)exp\left(j2\pi f_{d}t\right)$$
where A is the amplitude and $f_{d}$ is the Doppler frequency.

The received and transmitted signals are mixed with the original transmitted one, thus obtaining the low-pass filtered beat frequency signal
(3)$$S_{Rxm}\left(t\right)=Aexp\left(-j2\pi \left(\frac{Bt_{d}}{T_{c}}-f_{d}\right)t\right)exp\left(j\pi \frac{B{t_{d}}^{2}}{T_{c}}-j2\pi f_{0}t_{d}\right)$$
which is then sampled with the analog to digital converters (ADCs). The first exponential contains the target information of range and Doppler. The second exponential is a phase term which is not varying with time, it is constant across all fast Fourier transform (FFT) samples and it does not influence the range and Doppler calculation. By Calculating the FFT of the beat frequency signal the range response is derived. The complex FFT of Equation (Equation 3) is the spectrum of a truncated sinewave which is a sinc fucntion with −13.2 dB sidelobes level [13]. A second FFT, calculated per each range bin along the different chirps, extracts the Doppler induced frequency variation to measure the target velocity. Finally, if a MIMO array is exploited (not present in the system of this work), angle of arrival estimation is possible due to the known positions of the antennas.

The range resolution depends on the bandwidth of the transmitted signal $Rres=\frac{c}{2B}=\frac{c}{2ST_{c}}$ where *S* is the slope of the frequency modulated chirp. Once the sampling frequency ($F_{s}$) is fixed, also the maximum unambiguous range depends on the bandwidth $R_{max}=\frac{F_{s}cTc}{2B}$ (assuming in-phase (I) and quadrature (Q) sampling) and this is a limiting factor for parameters optimization.

Due to its nature of sampling beat frequency signals, the FMCW technology exhibits range and Doppler coupling. Target velocity affects range estimations as can be seen in Equation (Equation 3) and high Doppler frequency variations can degrade performances. Fast slope chirps can mitigate this effect at the cost of increasing the sampling frequency.

Another drawback is that the typical TDM MIMO scheme to achieve N transmitted signals orthogonality tends to limit the maximum unambiguous velocity since $V_{max}=\frac{\lambda }{4NT_{C}}$ where *N* is the number of transmitting elements. The ambiguous velocity can be recovered in different ways but this usually comes with severe drawbacks.

### 2.2. Binary-PMCW Radar

PMCW modulated waveform consists of periodically transmitting a certain number of sequences phase modulating a carrier frequency. By correlating the received and transmitted signals is possible to retrieve information on the range of the targets. Even if this modulated waveform is widely spread in communication and military radar applications, so far it has not been the main focus for automotive radars. The main reason is that, as opposed to an FMCW radar which sample the beat frequencies with a relatively low bandwidth (5–20 MHz typical for automotive applications), PMCW radar needs to sample the full bandwidth of the transmitted signal (1–2 GHz typical for automotive applications). For this reason, high rates sampling and accurate ADCs are required. With the advancement in chip integration, GSample per second ADCs can now be easily adopted. Another issue to be properly considered is the ADC dynamic range that can be harmonized with logarithmic amplifiers, high processing gains [14] and leaking cancellation techniques [15] to drastically improve sensitivity.

The most common PMCW modulated waveform is the binary PMCW which consists of a certain number of sequences of binary symbols $I_{n}$ (0,1) called chips with 0–π degree mapping of a carrier frequency. The transmitted sequence with a number of chips *N* and a chip duration $T_{ch}$ can be written as,
(4)$$s_{Tx}\left(t\right)=\sum_{n=0}^{N-1}g\left(t-NT_{ch}\right)cos\left(2\pi f_{0}t+I_{n}\pi \right)$$
where $f_{0}$ is the carrier frequency, $I_{n}$ is the sequence element and g(t) is a gate function of unit amplitude in the interval [0, $T_{ch}$]. The received PMCW signal after a propagation delay $t_{d}=\frac{2R}{c}$ can be modeled as
(5)$$s_{Rx}\left(t\right)=As_{Tx}\left(t-t_{d}\right)exp\left(j2\pi f_{d}t\right).$$

Calculating the correlation between $s_{Rx}$ and $s_{Tx}$, the range response is directly derived. An FFT calculated per each range-bin along the different sequences extracts the Doppler induced variation to measure the targets’ radial velocity. Also for PMCW radars, a MIMO approach (not present in the system of this work) is possible for angle of arrival estimation.

The range resolution is $Rres=\frac{T_{ch}c}{2}=\frac{c}{2B}$ where the difference with FMCW radar is that the bandwidth is driven by the duration of the chips.

Since, PMCW radar relies on sampling a time signal, the maximum unambiguous range is limited, in case of continuous transmission of the same sequence, by the duration of the sequence ($T_{s})$. Therefore, the maximum unambiguous range is $R_{max}=\frac{cT_{s}}{2}$ and it is independent of the bandwidth, providing a more flexible parameters optimization. In case of continuous transmission of orthogonal codes, unambiguous range can be extended.

Another advantage derived from the signal nature of the PMCW is that range-Doppler estimation is not coupled as shown in Equation (Equation 5).

With PMCW modulated waveform the adoption of orthogonal codes transmitted simultaneously by different antenna overcomes the limitation of TDM and drastically improves the limit on the maximum velocity.

As opposed to FMCW which present a sync function range compressed response when no window is applied, the autocorrelation of the PMCW sequences presents a thumbtack-like range response and sidelobes level can be significantly low with the right choice of the sequence allowing a much better HCR performance.

## 3. FMCW and Binary-PMCW Systems Description

### 3.1. FMCW

For the FMCW radar the well known AWR1843BOOST [16] evaluation board of the AWR1843 [11] radar SoC from Texas Instruments has been selected. The AWR1843 is an integrated single chip radar sensor based on the 45 nm radio frequency complementary metal-oxide semiconductor (RFCMOS) and operates in the 76–81 GHz automotive frequency band. To generate FMCW signals at the desired carrier frequency, an internal 40 MHz crystal oscillator followed by a clean-up phased locked loop (PLL) and a radio frequency (RF) synthesizer circuit are used. From the 20 GHz synthesized signal a 4× multiplier generates the proper signals for the transmitters and the receivers.

The RX chain has an RF noise figure of 15 dB and a phase noise at 1 MHz offset in the 77–81 GHz band of −93 dBc/Hz. Operations with 3 TXs and 4 RXs with I and Q channels are fully supported. The maximum output power of each TX channel is 12 dBm. The chip integrates an R4F ARM processor for automotive interfacing and a high performance C674x digital signal processor (DSP) for the radar signal processing.

The FMCW SoC samples the I and Q beat frequency signals with the internal ADCs. Range and Doppler FFTs are performed on the I and Q sampled data by the DSP and are stored in memory as a range-Doppler matrix which is then sent through the KSZ8851SNL [17] SPI-Ethernet converter to the laptop PC to be displayed on a user interface and stored for further analysis.

The antennas printed on the AWR1843BOOST print circuit board (PCB) are 3 elements vertical patch arrays with 10 dB gain at the design center frequency of 78 GHz. The calculated H–plane (horizontal) and E–plane (vertical) radiation patterns at 78 GHz are shown in Figure 1 and Figure 2 respectively. The horizontal 3 dB beamwidth is $\pm 28^{\circ }$, while the vertical 3 dB beamwidth is $\pm 14^{\circ }$.

### 3.2. Binary-PMCW

For the binary-PMCW radar a customized solution with 1 Tx channel and 1 Rx channel was been assembled. A block diagram of the prototype is shown in Figure 3 while the manufactured prototype is shown in Figure 4.

The transmitter is constituted by the evaluation board of the ADMV7320 [18] 81–86 GHz band upconverter from Analog Devices with typical saturation power Psat=26 dBm. The receiver is constituted by the evaluation board of the ADMV7420 [19] 81–86 GHz low noise down converter from Analog Devices with base band from DC to 2 GHz and a typical conversion gain of 10 dB. Between the receiving horn antenna and the ADM7420 evaluation board a Low Noise Amplifier (LNA), the AT-LNA-6090-1805T [20] from ATMicrowave, is present with a typical gain of 18 dB.

The evaluation board with the ADF5610 [21] chip from Analog Devices generates a 14 GHz reference signal for both the upconverter and the downconverter which is then internally multiplied by a 6x factor to obtain a 84 GHz carrier for the modulation.

The binary-PMCW modulating sequences are generated by the XC7K325T-2FFG900C FPGA hosted in the KC705 [22] development board from Xilinx. The sequences are translated into differential signals and sent to the in-phase upconverter input. The downconverted differential, I and Q, received signals are sampled by an AD9680 [23] chip hosted on a FMCDAQ2 board [24] from Analog Devices. The chip has 4 channels with 1 GSps and 14 bits precision and is connected to the FPGA through JESD204b [25] standard.

The FPGA receives the I and Q raw data sampled by the ADC, computes the correlations for the range processing and send the outputs through Ethernet to a laptop PC. The laptop PC computes the Doppler FFT in real-time to obtain a range-Doppler matrix which is displayed and stored for further analysis.

The antennas for the Tx and Rx channels are standard pyramidal horns. The antenna dimensions are 4.2 mm × 3 mm × 20 mm and it has a 10.5 dB gain. The H-plane (horizontal) and E-plane (vertical) radiation pattern at 84 GHz are shown in Figure 5. The horizontal 3 dB beamwidth is $\pm 20.5^{\circ }$, while the vertical 3 dB beamwidth is $\pm 21^{\circ }$.

## 4. FMCW and binary-PMCW Radar Parameters Description

### 4.1. FMCW

To achieve the comparison with the binary-PMCW system only 1 Tx and 1 Rx are enabled on the AWR1843 chip during the tests.

The FMCW frame is composed of 64 chirps with a duration ($T_{c}$) of 50 μs and with a bandwidth (*B*) of 250 MHz, and with a frame repetition rate ($F_{r}$) of 100 ms. The range resolution is $\delta R=\frac{c}{2B}=0.6$ m, where c is the speed of light. The number of samples of the range FFT for each chirp is equal to 256. Zero padding is performed to get 1024 range samples giving a range accuracy $R_{acc}=\frac{F_{s}}{2N_{FFT}S}=15$ cm, where $F_{s}$ is the sampling frequency equal to 7 MHz and *S* the slope of the chirps. The chosen parameters bring to a maximum unambiguos range $R_{max}=\frac{F_{s}cT_{c}}{2B}≃150$ m. The key parameters of the chirps and frame structure are reported in Table 1.

### 4.2. Binary-PMCW Complementary Code Sequences

The code sequences exploited by the binary-PMCW modulation are Golay complementary sequences [26]. This types of code is formed by complementary pairs which satisfy the properties of having out-of-phase aperiodic auto-correlation coefficients sum equal to zero [27]. Let a ($a_{0},a_{1},a_{2}\;\ldots \;a_{N}$) be a sequence of length *N* with $a_{i}\in \left\{+1,-1\right\}$ and its complementary pair b ($b_{0},b_{1},b_{2}\;\ldots \;b_{N}$) with $b_{i}\in \left\{+1,-1\right\}$ and define the aperiodic auto-correlation function as
(6)$$\rho_{a}\left(k\right)=\sum_{N=1}^{N-k-1}a_{i}a_{i+k},\;0\leq k\leq N-1$$

For the Golay complementary pair we obtain,
(7)$$\rho_{a}\left(k\right)+\rho_{b}\left(k\right)=0,\;k\neq 0.$$

This property is such that the sum of the responses of the complementary pair, transmitted one after the other, cancels out the sidelobes and doubles the peaks, resulting in a 2× improvement in SNR.

The sequences are generated by means of recursive construction. To generate the next code the previous code is concatenated to its complement. To generate the complement code the previous code is concatenated to the inverse of the previous code [28] as shown in Table 2.

### 4.3. Binary-PMCW

The binary-PMCW system generates Golay complementary sequences to modulate a carrier to be transmitted. On the receiver side, after downconversion, the I and Q signal are sampled. For each complementary sequence pair the correlation for a certain number ($N_{d}$) of delays (τ) between the transmitted sequence and the received I and Q signal is performed. The results are summed yielding a range response $R_{ab}\left(\tau \right)$
(8)$$R_{ab}\left(\tau \right)=\sum_{i=1}^{i=N_{d}}a\left(t\right)a_{rx}\left(t+i\tau \right)+\sum_{i=1}^{i=N_{d}}b\left(t\right)b_{r}x\left(t+i\tau \right).$$For each range bin a Doppler FFT is calculated obtaining a range-Doppler matrix.

The binary-PMCW frame is composed of 128 complementary sequences (64 pairs). Each sequence has 256 chips with a duration of 4 ns and with a bandwidth of 250 MHz. Also in this case the range resolution depends on the bandwidth and is equivalent to the FMCW one δR=0.6 m. The range accuracy depends on the ADC sampling frequency ($F_{s}$). Since $F_{s}=1$ GHz we obtain τ=1 ns and $R_{acc}=\frac{\tau c}{2}=0.15$ m. The a and b complementary sequences are transmitted continuously while, between the start of each pair, there is a 50 us blank time interval to obtain the same velocity resolution of the FMCW radar. The frame repetiton rate as for the FMCW is equal to 100 ms.

The key parameters of the sequences and frame structure are reported in Table 3.

### 4.4. Power Budget Analysis

To accurately compare the results, it was important to ensure that the power budgets of the two radars were the same so that the SNR was not affected by the hardware. Since the transmitted signal duration differed between the two radars, this needs to be considered when calculating the power budget. For this reason, the binary-PMCW system transmitter gain was set to obtain an output power $PTx_{P}=22$ dBm and the FMCW transmitter output power was set to be $PTx_{F}=10$ dBm. Since the gain of the transmitting antennas, the number of complementary sequences pairs, and the number of chirps were the same, the effect to be considered in the power budget for the different transmitted signal durations ($GTx_{t}$) was given by the ratio of the effective sampled time of a chirp ($T_{cs}=N_{ADC}/F_{S}=$ 35 μs) and the time duration of a complementary pair $T_{s}=2NTch=$ 2 μs
(9)$$GTx_{t}=10\log_{10}\left(\frac{T_{cs}}{T_{s}}\right)=10\log_{10}\left(\frac{35\;\mu s}{2\;\mu s}\right)=12\;\mathrm{dB}.$$With these settings, the relative power ratio ($R_{r}$) between the two system is 0 dB
(10)$$\begin{array}{c} R_{r}=\frac{PTx_{P}+GTx_{ant}}{PTx_{F}+GTx_{ant}+GTx_{t}}=\\ =\frac{22\;\mathrm{dBm}+10\;\mathrm{dB}}{10\;\mathrm{dBm}+10\;\mathrm{dB}+12\;\mathrm{dB}}=0\;\mathrm{dB}. \end{array}$$On the receiver side, the gains were set to get the same received signal strength and to comply with the ADC dynamic in both the systems. In this way, the binary-PMCW system LNA gain ($G_{LNA}$) together with the downconvertion gain ($G_{dc}$) achieved the same gain value of the FMCW internal LNA of the AWR1843:(11)$$\begin{array}{c} GRx_{P}=GRx_{ant}+G_{LNA}+G_{dc}=10\;\mathrm{dB}+18\;\mathrm{dB}+10\;\mathrm{dB}=38\;\mathrm{dB}\\ GRx_{F}=GRx_{ant}+GRx_{ch}=10\;\mathrm{dB}+28\;\mathrm{dB}=38\;\mathrm{dB}. \end{array}$$

## 5. Test Results

The two radar systems were tested simultaneously in an open grass field next to each other to capture the results from the same real-world scenario. The two radar are shown in Figure 6. On the left side, the two standard horn antennas of the binary-PMCW radar are visible on a wooden tile supporting the RF modules on the other side. On the right side the AWR1843BOOST evaluation board is clearly visible on the little wall.

As a first test, a 10 dBsm radar cross section (RCS) corner reflector was positioned at the boresight of the two radars at around 12 m distance. The measured range-Doppler matrices (on top) and the range responses (on bottom) of few consecutive frame are shown in Figure 7.

In the range response of the FMCW radar (Figure 7a) the typical −13 dB sidelobes of the peak can be observed. Instead, thanks to the Golay complementary sequences the sidelobes were cancelled out in the PMCW response (Figure 7b). To evaluate the effects of typical windowing, a Blackman window [29] with the frequency response parameters of Table 4 was applied (Figure 7c) to the range FFT inputs to reduce the sidelobes. While the sidelobes were effectively reduced, as a drawback the peak width was doubled and the maximum peak was reduced of 3 dB.

A test to verify the HCR capability of both technologies was performed by walking aside the corner reflector. Figure 8 shows the measured range-Doppler matrices (on top) and the range responses (on bottom) of few consecutive frames when the pedestrian was positioned near the corner reflector at a distance of 0.6 m. The binary-PMCW radar (Figure 8a) clearly separated the two targets presenting 15 dB difference in RCS (pedestrian typical RCS=−5 dBsm [30]) and corner reflector RCS=10 dBsm), when they were at a range difference ΔR≥0.6 m. The range resolution δR = 0.6 m limited further separation. For the FMCW radar (Figure 8b), despite having the same resolution δR = 0.6 m, the sidelobes range response of the corner reflector did not allow to separate the two targets. When Blackman windowing was applied to the range FFT input of the FMCW radar, despite the mitigation of sidelobes, since the peak width doubled, only a slight deformation was observable which was insufficient to separate the two targets (Figure 8c).

The tests result showed that the minimum distance at which the FMCW radar could separate the two targets is ΔR≥1.2 m even if the resolution was lower. In Figure 9 the results of the test when the pedestrian and the corner reflector were at ΔR=1.2 m are reported. Also in this case the binary-PMCW radar (Figure 9a) showed better performances separating clearly the two targets. The FMCW radar (Figure 9b) could barely separate the two targets but the detection was severely affected by the sidelobes. The FMCW radar with Blackman window (Figure 9c) revealed the small target but with a much worse SNR compared to the binary-PMCW radar.

The behavior of the two radars with respect to range was compared by detecting a walking pedestrian in front of the two radars. The pedestrian walked straight away from the radars, at their bore sights, at about 1.2 m/s, without changing speed and direction. In Figure 10, the range responses containing the maximum Doppler peak for each frame are reported, the two radars present identical performances and SNR as expected. In the AWR1843 Rx chain two first order internal high pass filters are applied to the analog beat frequency signal. This filtering improves the range dynamics attenuating strong RCS targets in close proximitiy of the radar. The two filters were set to the lower possible cut-off frequencies equal to 175 kHz and 350 kHz.

Tests to compare the Doppler responses were also performed moving a corner reflector toward and away from the radars. In Figure 11 the Doppler and range cut of the range-Doppler matrix for the position of the moving target are reported. Behavior of the Doppler response is identical for the two radars.

## 6. Conclusions

In conclusion, this paper presents and discusses a comparison of measured data in real-world scenarios, from a radar based on the currently most popular modulated waveform for automotive radar applications, the FMCW, and from a radar based on its main contender modulated waveform, the binary PMCW. Tests were performed on a grass field with an off the shelf FMCW radar and an assembled prototype binary PMCW radar in the same perfect conditions. The binary PMCW radar exploited Golay complementary sequences.

The binary PMCW modulated waveform enabled a reduction in the sidelobe levels and a drastically improved capability of separating weak RCS targets close to strong RCS targets. The results of the tests showed that the two modulated waveforms have identical performances in the Doppler response. Binary PMCW radars have some intrinsic advantages such as no range-Doppler coupling, shorter transmission time allowing for more transmitting antennas or larger unambiguous velocity and natural integration of communication and sensing. The only trade-off required to gain all these advantages is the use of higher sampling frequency ADCs that must be integrated into the binary PMCW radar chip.

The presented analysis demonstrates that the binary PMCW radar should be considered in the next generation of automotive radars.

## Figures and Tables

**Figure 1 sensors-23-05271-f001:**
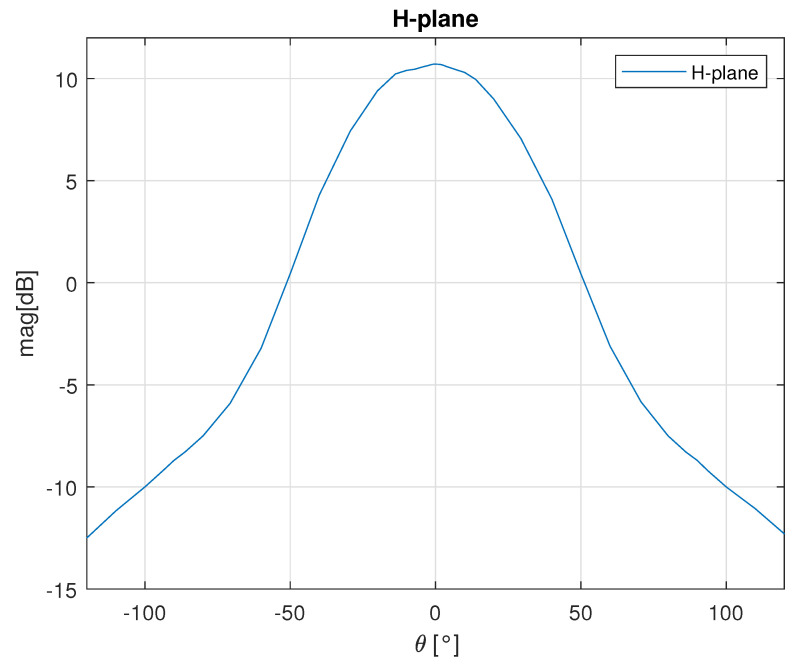
H-plane radiation pattern of the AWR1843BOOST.

**Figure 2 sensors-23-05271-f002:**
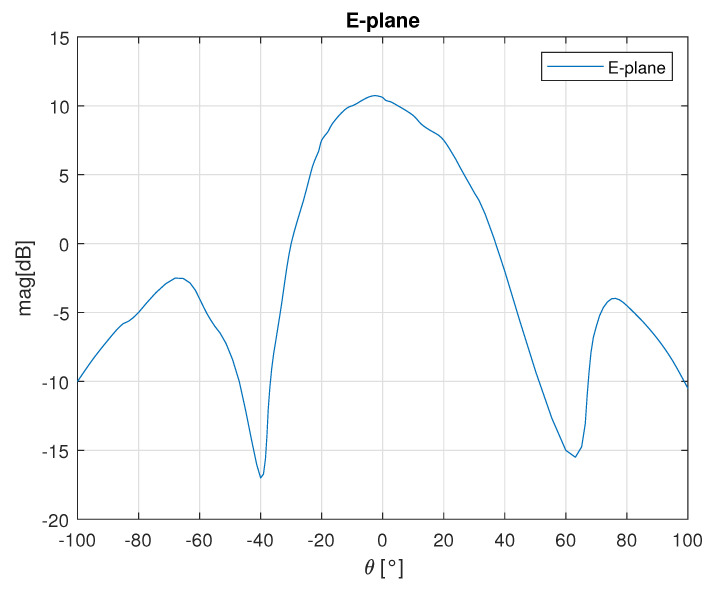
E-plane radiation pattern of the AWR1843BOOST.

**Figure 3 sensors-23-05271-f003:**
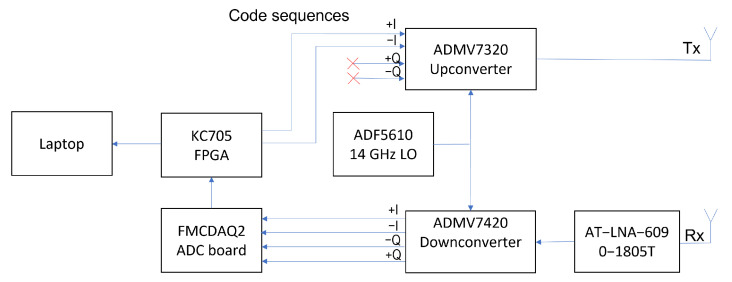
Block diagram of the binary-PMCW system assembled.

**Figure 4 sensors-23-05271-f004:**
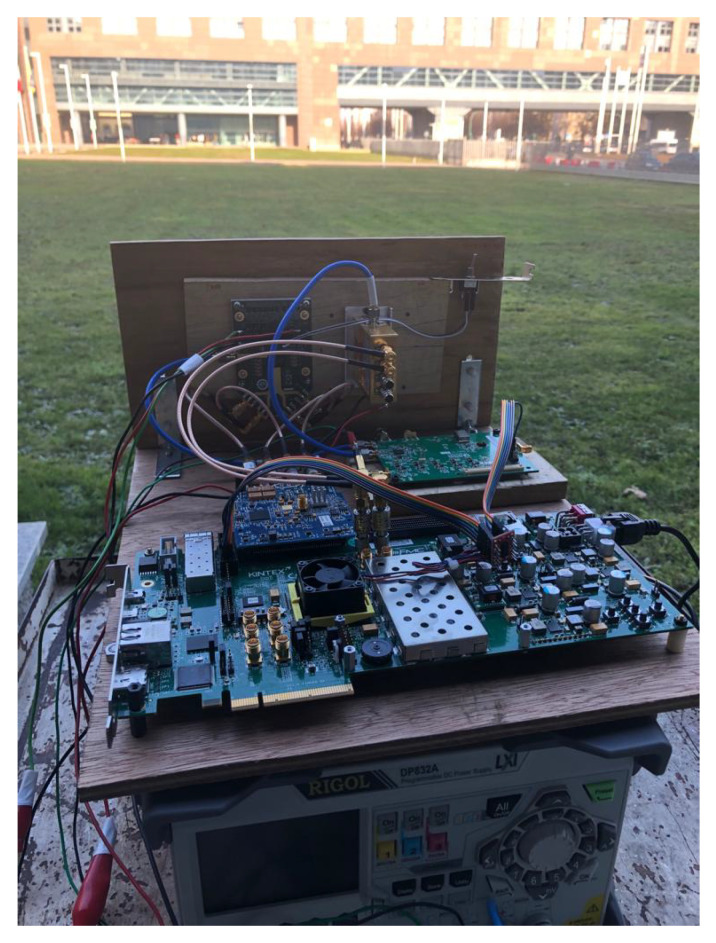
Binary-PMCW manufactured prototype.

**Figure 5 sensors-23-05271-f005:**
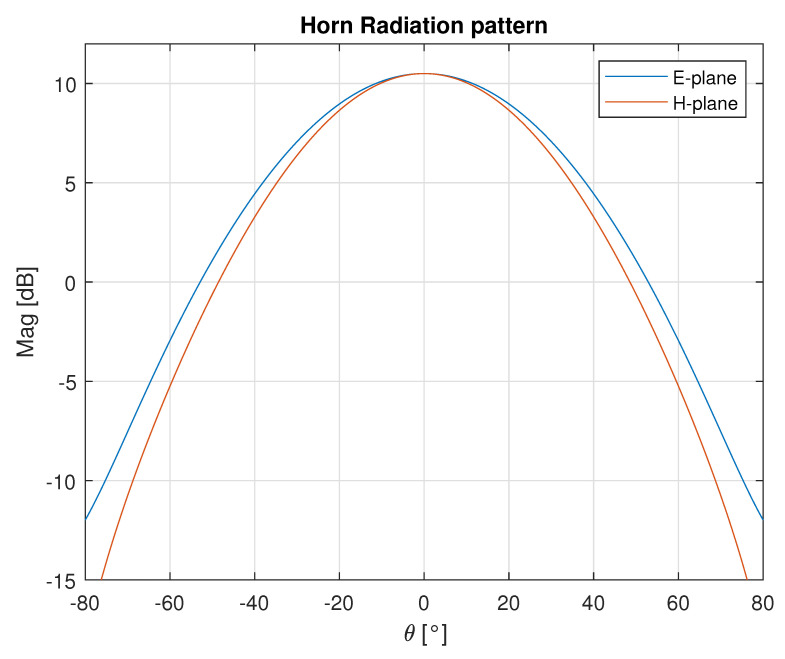
E and H plane radiation patterns of the standard pyramidal horn.

**Figure 6 sensors-23-05271-f006:**
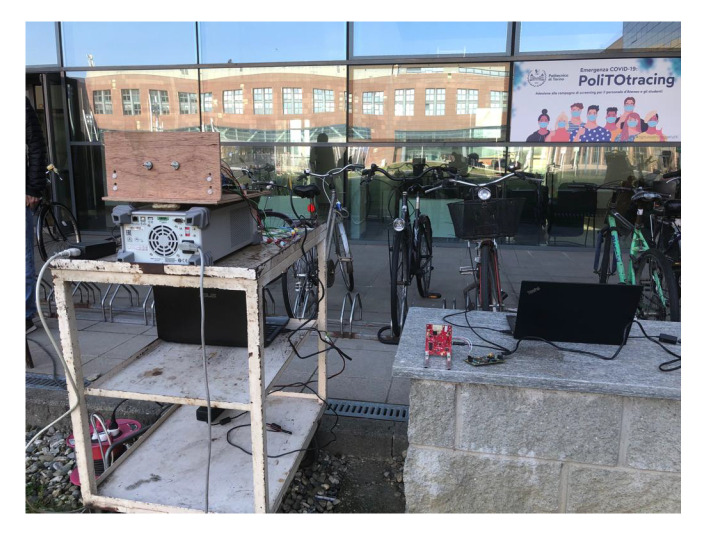
Tests Set-up.

**Figure 7 sensors-23-05271-f007:**
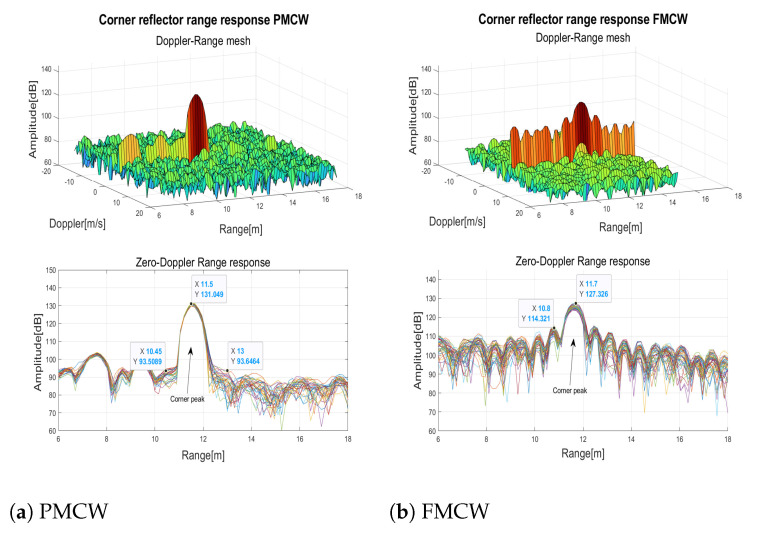
10 dBsm Corner reflector range response.

**Figure 8 sensors-23-05271-f008:**
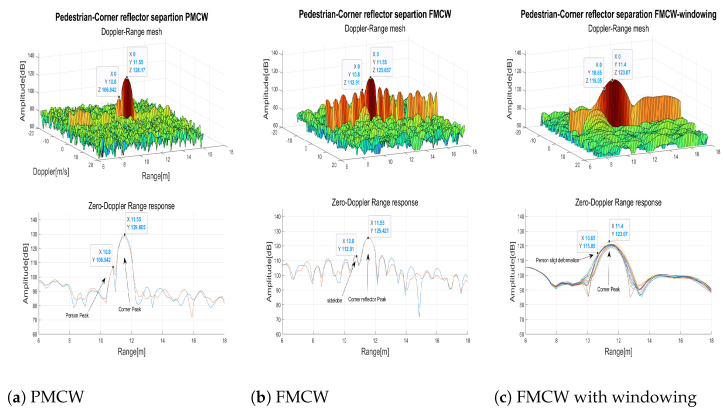
Separation of a pedestrian near to a corner reflector ΔR=0.6 m.

**Figure 9 sensors-23-05271-f009:**
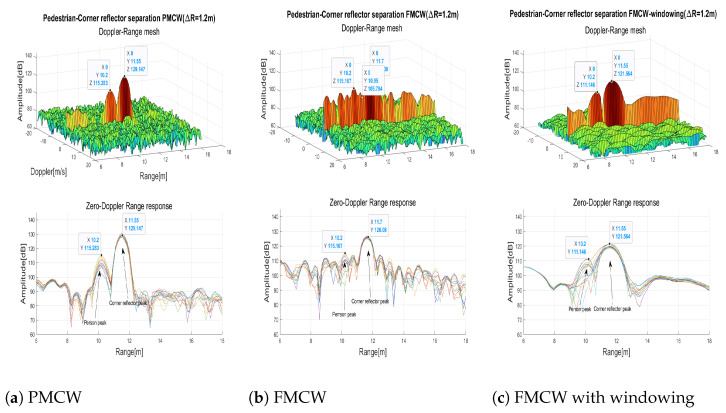
Separation of a pedestrian near to a corner reflector ΔR=1.2 m.

**Figure 10 sensors-23-05271-f010:**
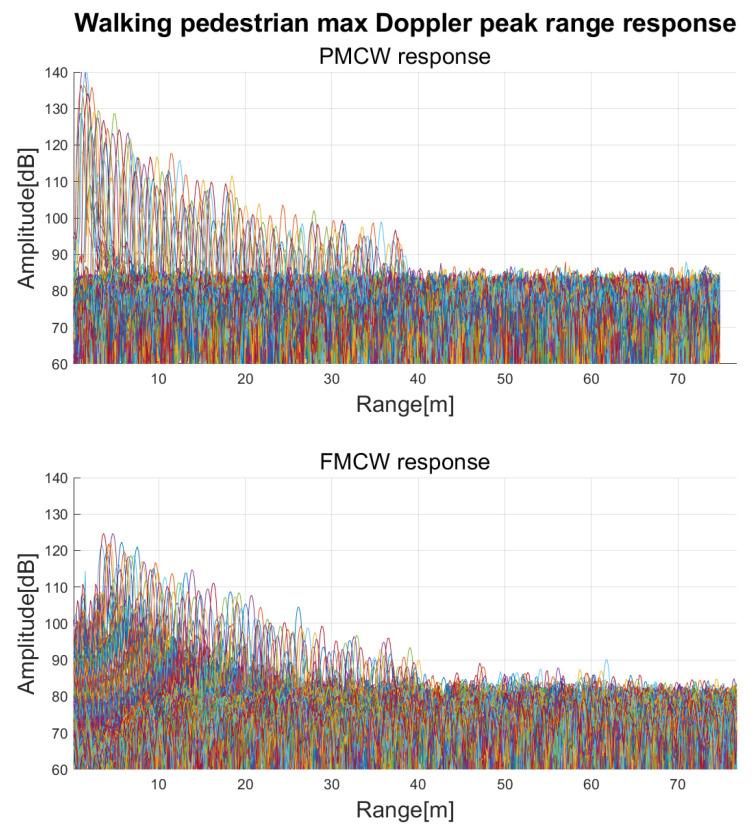
Maximum Doppler peaks of a walking pedestrian for each range.

**Figure 11 sensors-23-05271-f011:**
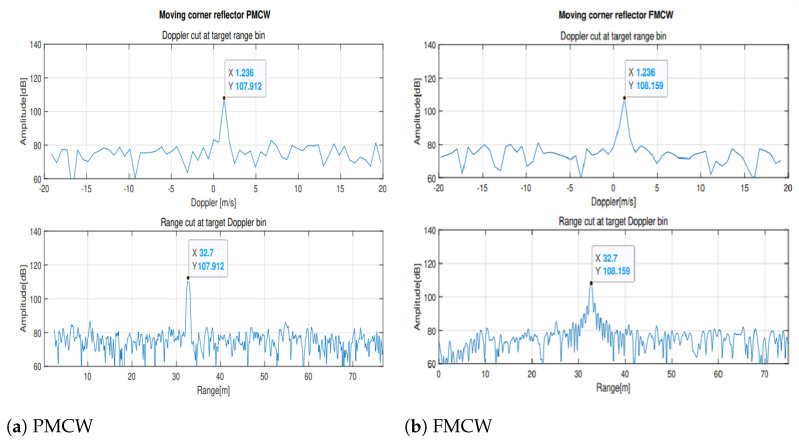
10 dBsm Moving corner reflector Doppler and range cuts.

**Table 1 sensors-23-05271-t001:** Key radar parameters for the FMCW frame and chirps.

Symbol	Description	Value
*B*	Bandwidth	250 MHz
$T_{c}$	Chirp Duration	50 μs
*S*	Slope	6.85 MHz/s
$N_{FFT}$	number of range FFT samples	1024
*N*	Number of chirps per TX	64
$F_{s}$	Sampling frequency	7 MHz
$F_{r}$	Frame repetition rate	100 ms
δR	Range resolution	0.6 m
$R_{acc}$	Range accuracy	0.15 m
$R_{max}$	Max unambiguous range	150 m
δV	Velocity resolution	0.6 m/s
$V_{max}$	Max unambiguous velocity	19.2 m/s

**Table 2 sensors-23-05271-t002:** Golay complementary sequences construction.

Code Length	Code	Complement
$2^{1}$	α	β
$2^{2}$	αβ	$\alpha \bar{\beta }$
$2^{3}$	$\alpha \beta \alpha \bar{\beta }$	$\alpha \beta \bar{\alpha }\beta$
…	…	…
$2^{N}$	$code_{N-1}compl_{N-1}$	$code_{N-1}{\bar{compl}}_{N-1}$

**Table 3 sensors-23-05271-t003:** Key radar parameters for the binary-PMCW frame and sequences.

Symbol	Description	Value
*B*	Bandwidth	250 MHz
$T_{ch}$	Chip Duration	4 ns
*N*	Number of chip in a sequence	256
$N_{p}$	Number of complementary pair	64
$F_{s}$	Sampling frequency	1 GHz
$F_{r}$	Frame repetition rate	100 ms
δR	Range resolution	0.6 m
$R_{acc}$	Range accuracy	0.15 m
δV	Velocity resolution	0.6 m/s
$V_{max}$	Max unambiguous velocity	19.2 m/s

**Table 4 sensors-23-05271-t004:** Blackman window frequency response parameters.

Description	Value
Max peak attenuation	3 dB
Sidelobes suppression	35 dB
Peak widening factor	2

## Data Availability

Data can be provided upon request to mattia.caffa@polito.it.

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
