# Peer review of "Binary-Phase vs. Frequency Modulated Radar Measured Performances for Automotive Applications"

_sensors, 2023, doi:10.3390/s23115271_

Round 1

Author Response

Dear reviewer,

Please see the attachment for the point-by-point response to the comments.

Kind Regards,

Mattia Caffa

Reviewer 2 Report

In this paper, the performance of FMCW and PMCW radar is compared on the measured data, which has a very practical significance. Review comments are as follows:
1 . The expressions for the Doppler frequency terms are inconsistent in equations (2) and (5) .
  • 2. Please use the subscript for "fd" in (3) .
  •  
  • 3.Please replace PCMW with PMCW for lines 117 and 122.
4. Please correct "rangee" in the line 290.

5. Please correct “therse is is a 50” in the line 236. 
  • 6. Please explain the difference between Rmax=Fs cTc/ (4B) in the line 93 and Ru=Fsc/(2S) in the line 207.
  •  

  •  
  •  

Author Response

Dear reviewer,

Please see the attachment for the point-by-point response to the comments

Kind Regards,

Mattia Caffa

Reviewer 3 Report

This paper investigates comparisons of measured performances between a 3 TX/4 RX FMCW radar and a 1 TX/1 RX Binary-PMCW radar in real-world conditions environment. I think that the presented idea is interesting and the manuscript is generally well written. However, there are still some problems as specified below. The authors are recommended to address and fix the them before resubmitting.

Major observations:

1.      In Section 5, The authors are advised to explain what is a scenario of a walking pedestrian in front of the two radars in Figure 10. For example, the authors should give the gait of a walking pedestrian.

2.      There is no simulation reflecting that the Binary-PMCW radar can obtain larger unambiguous velocity.

3.      I think that interference mitigation for both FMCW and PMCW should be considered, since it is a very important issue in automotive application. The following closely related literature should be discussed.

[R1] Liu, C.; Liu, S.; Zhang, C.; Huang, Y.; Wang, H. Multipath propagation analysis and ghost target removal for FMCW automotive radars, In Proceedings of the 2020 IET International Radar Conference (IRC), 2020, pp. 330–334. https://doi.org/10.1049/icp.2021.0554.

4.      Both advantages and disadvantages of the Binary-PMCW radar should be comprehensively discussed in the discussion and conclusion section. Is there any cost that must be paid to gain the advantages?

Minor observations:

1.      In Section 2,the Figure 1 and Figure 2 are not very clear.

2.      In Section 4,$ R_{ab}(i\tau) $ should be $ R_{ab}(\tau) $ in Equ (8).

3.      There are several editing errors in Equ. (8), Figure 6 (a) PMCW, Line 236 and Line 290. The authors are advised to check the manuscript thoroughly to reduce such errors.

Author Response

(The authors gave the same response as above.)

Reviewer 4 Report

Contribution:

The authors assembled the prototype to realize the binary-PMCW radar and compared it with the FMCW radar in real environment. This has a very good contribution and potential for PMCW radar research. 

Major issues:

1. This paper points out the limit of maximum velocity of the FMCW radar when using TDM, but it does not seem appropriate because the two systems were compared using the 1Tx 1Rx system. Could you supplement this part?

2. The FMCW signal description in (2.1) seems insufficient. There is no carrier frequency term in (eq 1) and insufficient explanation of the reason why the second exponential term is not related to the target's information (84 - 86 lines). 

3. In the block diagram of (Figure 3), could you explain why there are seperated +-  line of In-phase and Quadrature?

4. The author compared the PMCW radar with FMCW radar using blackman window in the (5. test results). There is lack of explanation why you chose this blackman window instead of other windows. It is necessary to state why you chose the window, or you have to compare with other windows such as Hamming, Bartlet window, etc.  

5. It is better to explain mathematically why the sidelobe level of the FMCW radar is larger than binary-PMCW through modulated waveform design and how the binary-PMCW modulated waveform allows a reduction in the sidelobe level. By doing so, authors can make a more convincing conclusion that PMCW radar should be adopted in the automotive radar system for next generation.

6.  It seems necessary to supplement the description of the internal high pass filter of the FMCW radar in lines 293 to 295 of (Figure 10). Could you explain the step when internal high pass filter is applied and why this filter is applied to the FMCW radar?

Minor issue:

1. The letters in figures are small. 

2. In Figure7-(a) PMCW range response, the title should be PMCW, not FMCW

3. typos in mathematical expression and words.

A moderate modification of English composition is needed. There are some typos in English words and awkward sentences.

Author Response

(The authors gave the same response as above.)

Round 2

Reviewer 1 Report

The manuscript bearing title “Binary-Phase vs Frequency Modulated Radar Measured Performances for Automotive Applications” is accepted for the publication in the present form. All the comments are well-addressed. I think this paper is ready for publication.

Perfect

Reviewer 3 Report

I think that the manuscript in its present form is ready for publication.